# Digitally Inclusive, Healthy Aging Communities (DIHAC): A Cross-Cultural Study in Japan, Republic of Korea, Singapore, and Thailand

**DOI:** 10.3390/ijerph19126976

**Published:** 2022-06-07

**Authors:** Myo Nyein Aung, Yuka Koyanagi, Yuiko Nagamine, Eun Woo Nam, Nadila Mulati, Myat Yadana Kyaw, Saiyud Moolphate, Yoshihisa Shirayama, Kumiko Nonaka, Malcolm Field, Paul Cheung, Motoyuki Yuasa

**Affiliations:** 1Department of Global Health Research, Graduate School of Medicine, Juntendo University, Hongo 2-1-1, Bunkyo Ku, Tokyo 113-8421, Japan; y-koyanagi@juntendo.ac.jp (Y.K.); m.nadila.vp@juntendo.ac.jp (N.M.); myat.rr@juntendo.ac.jp (M.Y.K.); shirayam@juntendo.ac.jp (Y.S.); moyuasa@juntendo.ac.jp (M.Y.); 2Advanced Research Institute for Health Sciences, Juntendo University, Hongo 2-1-1, Bunkyo Ku, Tokyo 113-8421, Japan; 3Faculty of International Liberal Arts, Juntendo University, Tokyo 113-8421, Japan; 4Department of Judo Therapy, Faculty of Health Sciences, Tokyo Ariake University of Medical and Health Sciences, Tokyo 135-0063, Japan; 5Department of Family Medicine, Tokyo Medical and Dental University, Tokyo 113-8510, Japan; yuiko.mail@gmail.com; 6Department of Health Administration, Software Digital Healthcare Convergence College, Yonsei University, Wonju 26493, Korea; ewnam@yonsei.ac.kr; 7Department of Public Health, Faculty of Science and Technology, Chiang Mai Rajabhat University, Chiang Mai 50300, Thailand; saiyudmoolphate@gmail.com; 8Tokyo Metropolitan Institute of Gerontology, Tokyo 173-0015, Japan; nonaka@tmig.or.jp; 9Faculty of Social Sciences, Kyorin University, Tokyo 181-8611, Japan; marukomu@ks.kyorin-u.ac.jp; 10Faculty of International Liberal Arts, Waseda University, Tokyo 169-0051, Japan; 11Asia Competitiveness Institute (ACI), Lee Kuan Yew School of Public Policy, Singapore 259772, Singapore; paul.cheung@nus.edu.sg

**Keywords:** healthy aging, gray digital divide, digital inclusion, empowerment, cross-cultural, Asia, mixed method

## Abstract

One out of three people in Japan will be an older person before 2040. Half of those currently do not utilize the internet, smartphone apps, or digital technology. On the other hand, more than 70% of seniors in Republic of Korea use the internet, and 55% in Singapore had access to it in 2019. The use of digital technology for health promotion has the potential to promote individual and community empowerment, advocating for healthy, active aging. Maintaining equity in health promotion practice requires the digital inclusion of every senior. Therefore, we propose a cross-cultural study to explain the contextual influences of digital inclusion and its consequences on healthy aging in Japan, Korea, Singapore, and Thailand. Quantitatively: digital skills, e-health literacy, participation in health promotion, and quality of life will be analyzed in structural equation models. Qualitatively: thematic analysis will be developed to identify cultural patterns and contextual factors, making sense of what older persons in different countries materialize, say, do, think, and feel to reveal deeper beliefs and core values about digital inclusion and healthy aging. Logics and methods from this protocol would be useful to replicate the study in many countries globally. Evidence from this study is expected to pave the way to digitally inclusive, healthy aging communities (DIHAC) across Japan and Asia.

## 1. Introduction

Aging populations have become a global phenomenon. Asia has more than 600 million older people, which is 57% of the world’s older population [1] and is the largest number by region. Super-aging Japan, having nearly 30% of its population aged over 65 years, is the oldest in Asia, followed by other fast-aging countries: Korea (25%), Singapore (20%), and Thailand (20%). By 2030, the retirement age in those countries is expected to be raised from 65 to 70 in Japan and from 60 to 65 in Singapore to stabilize social security revenue [1]. Therefore, healthy aging and the productivity of the mature workforce have become central to sustainable productivity. These two targets can be addressed by digital technology [2]. Digital technology has an indispensable role as an enabler of a healthy aging society. Simply, the use of digital technology can contribute to the well-being of older adults and empower them to age with dignity [3]. Previous studies in Asia reported frequent use of the internet and its benefit for physical, mental, and social well-being [4]. It has been reported that older people who live alone and have limited engagement in work and social activities felt more socially connected after using digital technology, showed improvement in their well-being, and the feeling of loneliness and social isolation had reduced [5,6]. The use of information and communication technologies (ICTs) among older is related to greater preventative health behaviors [7] and improved cognitive function [8]. Moreover, the use of digital technologies among the oldest old (80+) has shown a positive effect on their subjective well-being domains of loneliness, anomie, and autonomy [9].

Consequently, digital transformation policy is increasingly fostered by governments in Asia, especially Japan, Korea, Singapore, and Thailand. From this political viewpoint, digital technology has become the most significant transformative agent to maximize older adults’ independence, health, and productivity.

Achieving healthy aging requires the adoption of holistic approaches to address inequalities to improve social determinants of health [10]. The inequality that is unique to the older persons and urgent to address is the “grey digital divide” [11]. Unlike the Millennium generation, baby boomers, the older generation, did not get the opportunity to learn to use digital technology and media in basic education. The 2019 Fundamental Rights Survey in the European Union shows that only one in five survey respondents (20%) aged 75 and older at least occasionally engaged in internet activities, compared to 98% of those aged 16–29 years [12]. Although many older people in developed countries now own or have access to a computer, smartphone, or internet connection and use social media, their internet and digital usage varies across cultures. Whereas the internet penetration rate for the general population is consistent at around 90% in Japan, Korea, and Singapore [11] and 88.6% in Thailand [13]. In Japan, more than 90% of the individuals aged 13–59 used the internet in 2020, whereas this proportion was 59.6% and 25.6% for individuals aged 70–79 and aged 80 and over, respectively. For Singapore, the rates for individuals aged less than 15 (90%), 15–24 (100%), and 25–74 (94%) made up the biggest portion of internet use in 2020, whereas only 46% of the older adults (age 75 and over) used the internet in 2020 [14]. Moreover, 76.6% of Korean older persons were internet users, whereas general public internet usage is 91.1% in 2020 [15]. According to the national survey of Thailand, 81.8% of Thai population used the internet in 2021, while those aged over 50 years constituted less (61.1%) [13].

COVID-19 has accelerated the use of digital technology in different areas, from information dissemination and tracking of new transmission [16] to provision of social connection during lockdowns [17]. A new normal lifestyle with physical distancing measures limits group activities such as community group exercise, gathering, and civic participation, therefore weakening the social connections and causing older people to be prone to social isolation. It is observed that physical inactivity is increased in both robust and frail older people [18]. Nowadays, mobile applications and virtual and web-based environments are progressively being used to organize health promotion and socializing activities, replacing traditional community group exercise and social and cultural events to counteract the negative effect of social distancing. These strategies have shown effectiveness in improving physical activity [19] and reducing loneliness [20]. Therefore, it is important to curtail the digital divide among the seniors and foster digital inclusion for healthy aging, taking different cultural contexts into account. Current Japanese policies are targeting, promoting access to the internet among seniors to minimize the grey digital gap. Evidence-based empowerment of the individuals and communities is required to enhance the impact of this policy. Therefore, it is important to understand the social factors such as family support, peer support, and cultural patterns fostering older persons’ acquisition of digital skills and their use for daily life and healthy aging.

Digitally active seniors can stay connected, prevent social isolation, and can access online health resources and services. However, there is a significant gap in the adoption and diffusion of digital technology among older persons, which varies across different cultures [21]. What makes the difference in the adoption of digital technology among seniors? How can we empower older persons to be digitally inclusive and get healthy aging advantages? No study has answered these critical questions to date. Therefore, we propose a cross-cultural study to explain the contextual influences of digital inclusion and its consequences on healthy aging in Japan, Korea, Singapore, and Thailand. Our research questions are: (1) What are the contextual influences for digital inclusion? (2) What are its consequences on healthy aging in Japan, Korea, Singapore, and Thailand? To answer these research questions, we developed a study protocol and shared the logic and methods for replicating further studies globally.

The main objective of this study is to identify contextual influences of digital inclusion and its consequences on healthy aging in Japan, Korea, Singapore, and Thailand. The specific objectives of this research are (1) to measure and comparatively assess the digital skill of the older persons across four countries, (2) to measure and compare the e-health literacy of older persons in four countries, (3) to identify causal and sequential factors of digital inclusion among older persons such as personal characteristics, technical characteristics, and positional characteristics, (4) to identify paths and association between digital inclusion and older persons’ participation in health promotion and health-related quality of life, and (5) to exchange the contextually grounded models leading to digital inclusion and its impact on senior’s healthy aging in four countries.

## 2. Research Methods

### 2.1. Study Design and Sampling

This is a five-year project. A mixed-method study will apply an explanatory sequential (QUAN → qual), two-phase research design [22]. In each country, older participants will be recruited in a city that has ongoing health promotion activities for the seniors. We estimated the sample size for quantitative to have a power of 80% and 95% confidence interval for the finding. Estimated samples constitute 360 participants for surveys in Japan, 360 in Korea and 480 in Thailand, and 360 in Singapore. Sample size calculation applied one sample estimation of proportion in Stata SE16, based on the percentage of internet use among the older people. For instance, internet use among the older people in Japan has been reported as 50% (null), but we estimated it as 40% (alternative), and we applied the command for one sample estimation in StataSE16. We aimed to get a sample sufficient to analyze small town models applying structural equation modeling. In addition, we also checked to secure the sample to exceed ten times the factors, less than 30, in the instrument number for each factor. Generally, the number of each sample was inflated by 20% for compensating the non-responses.

The inclusion criteria of the study participants will be the older residents (aged 65 years and over, or as nationally defined) of the community that has ongoing community-based health promotion activities for seniors; both males and females will be included in the study.

### 2.2. Study Phases and Research Instruments

#### 2.2.1. Phase One: Quantitative Survey

Objectives of the study phase one comprise (1) to measure and comparatively assess the digital skill of the older persons across four countries, (2) to measure and compare the e-health literacy of older persons in four countries, (3) to identify causal and sequential factors of digital inclusion among older persons such as personal characteristics, technical characteristics, and positional characteristics, and (4) to identify paths and association between digital inclusion and older persons’ participation in health promotion and health-related quality of life.

#### Research Instruments

Translation of instrument will follow forward translation, backtranslation, cognitive test, and pilot study to check the reliability and retain the validity. The research instruments that will be used in this study are as below:Digital Skill Measurement Scale

The perceived digital skill of the study participants will be measured using the “Digital skill measurement scale” of the London School of Economics, transcultural translated version [23]. It is a 23-item, 5-point Likert scale measuring five domains: operational internet skill, information navigation, communicational/social internet skill, creative skill, and mobile internet skill.

2.eHealth Literacy Scale

Norman and Skinner defined the concept of eHealth literacy as the ability to seek, find, understand, and appraise health information from electronic sources and apply the knowledge gained to addressing or solving a health problem [24]. The participants’ perceived skill at using information and communication technologies for health will be assessed using an 8-item 5-point Likert eHealth Literacy Scale [25].

3.Health Promotion Activity Participation Registers or Record

The study participants will be selected from the community that has ongoing health promotion activities for older adults. Therefore, the registration record will be obtained with the permission of the participants and activity organizers. The type and frequency of the activity will be specified.

4.ICECAP-O (ICEpop CAPability measure for Older people)

In this study, we focus on community-dwelling older adults and their participation in health promotion activities. In measuring the quality of life, health is an important domain but not the only one. Capability measures the ability to do valuable things or be with the given resources. According to Sen’s capability approach in measuring the quality of life, measuring the quality of life should not be only based on perceived satisfaction but also include the conditions that allow or enable oneself to do things or be that one values [26]. Capability also differs according to an individual’s characteristics and the social environment that they interact with. Therefore, we will use ICECAP-O Instrument for older people to measure the capability and well-being of study participants. It is a 5-item tool that measures capability and the extent a person can do things that they value [27] and reflects the general quality of life in a broader sense [28]. The domains of ICECAP-O, based on the capability concept, include (1) attachment, (2) security, (3) role, (4) enjoyment, and (5) control [29]. The domains reflect the essence of healthy and active aging. It coincides with the study’s aim to explore the effect and sequences of using digital technology on the healthy aging of the older population.

5.Personal characteristics, technical characteristics, and positional characteristics, family factors, peer factors, and community factors will be investigated by applying resource and appropriation theory in digital sociology [11].

#### 2.2.2. Phase Two: Qualitative Inquiries

The second phase of this study is qualitative inquiry, which aims to identify the cultural pattern, and contextual factors, making sense of what older persons in different countries materialize, say, do, think, and feel to reveal deeper beliefs and core values about digital inclusion and healthy aging.

Moreover, we seek to exchange the contextually grounded models leading to digital inclusion and its impact on seniors’ healthy aging in four countries.

At least five focus group interviews inviting older persons, family caregivers, and multiple stakeholders will be conducted at each site. Data collection will also apply field observation, voice recording, photography and digital videos to construct the empowerment models through socio-cultural and environmental factors, social interaction, values, challenge and adaptation of older persons in each setting.

##### Research Instrument

Qualitative thematic analysis will identify the cultural patterns and contextual factors, making sense of what older persons in different countries materialize, say, do, think, and feel to reveal deeper beliefs and core values about digital inclusion and healthy aging, applying a cross-cultural empathy framework [30] (Table 1).

Findings from the two phases will be critically reflected and integrated to identify the DIHAC model in each country and for Asia in the study year 4 (Table 1). Applying a cross-cultural empathy framework, this explanatory sequential two-phase research may chart the Asia DIHAC model to strategize the empowerment of communities and individuals for healthy aging communities.

### 2.3. Ethical Consideration

The Juntendo University Ethics Committee has reviewed and approved the protocol of “Digitally Inclusive, Healthy Aging Communities (DIHAC): A Cross-Cultural Study in Japan, Korea, Singapore, and Thailand” (DIHAC) study. The approval number is E22-0057-M01.

Written informed consent will be obtained from all participants, and the purpose of this study will be carefully explained to them after their permission.

### 2.4. Data Analyses

Stata SE 16 (Stata Corp 4905, Lakeway Drive, College Station, TX, USA) will be applied for data analysis. We will use descriptive analysis to elaborate on the sociodemographic information of study objectives. Statistical analysis of various factors will apply bivariate analysis and multi-variable regression model. A *p*-value of less than 0.2 in the bivariate analysis or logical relevance will be certified as the factor to be selected in the multivariable regression analysis. The structural equation modeling (SEM) method will be used to detect pathways from explanatory variables to the outcome variables, constructing the conceptual model or mechanism. It identifies the unobserved variables. It can reveal the direct and indirect association. Furthermore, it allows us to see not only the causal pathways but also consequential pathways [31]. After we develop the SEM models, model fit indexes will be computed. Statistical significance is defined as a *p*-value of less than 0.05 with a 95% confidence interval (CI).

For the qualitative analysis, we will use a cross-cultural empathy framework to carry out inquiries and analyze the collected data. Literature review to understand the development of digital policies and healthy aging programs in four countries and to understand the systems and contextual factors will precede the data collection and analysis. Cross-cultural exchange in the results and model aims to identify how digital policies are being adopted by the older persons, facilitators, barriers in each setting, and, finally, the empowerment models. 

## 3. Expected Result and Discussion

Through our cross-cultural study, we hope to identify contextual influences of digital inclusion and its consequences on healthy aging in Japan, Korea, Singapore, and Thailand. To date, we have not identified a previous study that has been conducted in this area of research.

In 2020, the COVID-19 pandemic caused the suspension and cancellation of group exercise activities and social and cultural activities for the older adults in the communities in Tokyo. Researchers from the DIHAC team collaborated with the primary health care providers in a community in Tokyo. The objective was to sustain health promotion activity by introducing telehealth education and a home exercise program for seniors [32]. Video conferencing platforms such as Zoom could connect members of community group exercises. However, participation is still few because of the gap in usage of the internet, mobile apps, and social media. The researchers posited that if they could improve the aging population’s usage of the internet and digital technology, they would be able to sustain their active social network and civic participation, and health promotion. To achieve this goal, we would need to improve the digital skills of the seniors in Japan.

Recently, the use of the internet among older persons rapidly increased up to 70% in Korea, according to national data [15]. Likewise, 58% of Singaporean seniors aged above 60 used the internet mostly for messaging, socializing, and information retrieval [33], whereas Thai seniors increasingly embrace Mobile Health (m-Health), which is a component of eHealth. The Global Observatory for eHealth defined mobile health (mHealth) as medical and public health practice supported by mobile devices, such as mobile phones, patient monitoring devices, personal digital assistants (PDAs), and other wireless devices and its benefits. The knock-on effect for societies would be that more job opportunities would be available if healthy aging seniors can use the internet and digital technology.

Health promotion practices usually address inequality in social determinants of health (SDH), such as education, economic status, urban or rural residence, and social environment. Digital transformation contributes to improving the SDH of the older persons, such as their social wellness through being connected, their financial wellness through employment opportunities for digital skills, and their health through e-health literacy and informed choices, digital health promotion, and telehealth services. Therefore, digital inclusion is a core contributing factor to those SDHs [1,2]. The “Gray digital divide”, a digital divide among seniors of age 65+ years, is the most serious gap or new inequity that we need to address as a social determinant of health for the older population.

We know that internet use and digital technology would enable seniors for healthy, active, and productive in later life. The barrier is the digital divide. Therefore, we view the digital divide as inequality in the determinant of health, which remains to address in contemporary aging Asia and globally. We view digital inclusion as pluripotential contribution factors to other social determinants of health (SDH). Recently, the Japanese government invested more to secure internet access for the seniors. Timely research is required to identify the strategy to promote older persons’ “digital usage” empathetically. We still need research that will find out the factors which can enable the seniors to use digital technology, interacting and mobilizing social resources such as family, community, peers, personal characteristic, technical characteristic, and positional characteristic. Hence, a cross-cultural explanatory study is worth exploring the models in Japan, Korea, Singapore, and Thailand and integrating the strength and uniqueness of different models. As a result, the findings of this research will strategize the empowerment of communities and individuals for co-creating digitally inclusive, healthy aging communities.

## 4. Conclusions

With the policies of governments promoting digital inclusion, we learned that internet access has been increasingly secured. The gap remains in the usage of the internet and digital technology among older adults. To bring about healthy aging, it is necessary to empower the older person to be digitally inclusive. The literature review identified that the empowerment of individuals and communities is a common strategy for both health promotion and digital inclusion [10,11]. Theoretically, it is clear. However, social factors such as cultural patterns and contextual influences remained unclear on how to maximize the digital inclusion of the seniors in the Asian context. DIHAC study team combines the knowledge, evidence, and experiences through bi-monthly organized, international DIHAC policy review meetings, which constitute a platform for cross-cultural exchange among researchers from different countries and an opportunity to develop young researchers [34]. Ultimately, the findings of this study would be useful as an empowerment strategy and evidence-based guidance to pave the way to digitally inclusive, healthy aging communities (DIHAC) across Asia and the world.

## Figures and Tables

**Table 1 ijerph-19-06976-t001:** What we will elucidate in four study sites and the expected yearly outcome in five-year timeline.

Timeline	Japan	Korea	Singapore	Thailand
**Year 1**	Coordination meeting in Zoom	Coordination meeting in Zoom	Coordination meeting in Zoom	Coordination meeting in Zoom
	Policy review	Policy review	Policy review	Policy review
	Preparation of instrument, validation, and translation	Preparation of instrument, validation, and translation	Preparation of instrument, validation, and translation	Preparation of instrument, validation, and translation
	Ethical approval	Ethical approval	Ethical approval	Ethical approval
	Coordination visit	Coordination visit	Coordination visit	Coordination visit
Output	A detailed plan and ethical approval in hand and launching the DIHAC website digital-ageing.com			
**Year 2**	Filed survey: Quantitative Data collection	Filed survey: Quantitative Data collection	Filed survey: Quantitative Data collection	Filed survey: Quantitative Data collection
	Statistical analysis	Statistical analysis	Statistical analysis	Statistical analysis
Output	Preliminary data analysis results presented in Japan or international conference			
**Year 3**	Interview visit	Interview visit	Interview visit	Interview visit
	Qualitative data collection	Qualitative data collection	Qualitative data collection	Qualitative data collection
	Thematic analysis	Thematic analysis	Thematic analysis	Thematic analysis
	Data integration (QUAN → qual)	Data integration (QUAN → qual)	Data integration (QUAN → qual)	Data integration (QUAN → qual)
Output	Presenting qualitative results in Japan or international conference			
**Year 4**	Model synthesis: Cross-cultural critical reflection via meeting/workshop/Zoom meeting			
	Japan model	Korea model	Singapore model	Thai model
Outcome	**Asian model of DIHAC and empowerment strategy**			
Output	Presenting at an international symposium			
**Year 5**	Dissemination of DIHAC study findings			
	Preparing initial publication to submit to peer-review international scientific journals			
Outcomes	Preparing a report, final manuscripts and a book, videos of case studies, visual media			
Outcomes	Advocacy: informing results to the local authority, reporting models, and empowerment strategy			
Outcomes	Publish manuscript	Publish manuscript	Publish manuscript	Publish manuscript
	**Final DIHAC symposium in Japan**			

Note: (QUAN → qual)—quantitative findings and qualitative findings are integrated to explain the results; DIHAC—digitally inclusive, healthy aging communities.

## Data Availability

The study is in the preparation phase for data collection. Sharable data are in the public access at DIHAC study website (digital-ageing.com) as regular meetings reports, videos, photos, news, press release and publications. [34] Annual study progress reports are accessible on Kakenhi page for DIHAC study [35].

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
