# Peer review of "Digitally Inclusive, Healthy Aging Communities (DIHAC): A Cross-Cultural Study in Japan, Republic of Korea, Singapore, and Thailand"

_ijerph, 2022, doi:10.3390/ijerph19126976_

Round 1

Reviewer 1 Report

Thank you for the opportunity to review this study protocol on the contextual influences of digital inclusion and its consequences on healthy ageing in Japan, ROK, Singapore, and Thailand.

General comments:

It would be useful to do a systematic review of the literature on this topic or at least a scoping review prior to your study.

Introduction:

  1. Lines 68-70: Could you confirm the meaning of this sentence? I have the impression that there is a misunderstanding.
  2. Lines 113-120: Could you rephrase the objective sentence to select only one of the objectives as primary objective and define the others as secondary objectives?

Methods:

  1. Lines 123-125: Could you provide more information on the sample size calculation?
  2. Lines 125-127: The inclusion criteria of the patients (age, cognitive ability, visual and auditory ability,...) should be explicitly mentioned.
  3. Lines 127-128: Could you provide the reference of the “Digital skill measurement scale” questionnaire and a copy of this questionnaire in the appendix?
  4. Lines 128-129: Could you define the "eHealth Literacy" concept?
  5. Lines 128-129: Could you provide a copy of the “eHEALS eHealth Literacy Scale” questionnaire in the appendix?
  6. Lines 130-132: Could you provide a copy of the “ICECAP-O (ICEpop CAPability measure for Older people)” questionnaire in the appendix?
  7. Lines 130-133: Could you give examples to illustrate that the capability reflects a broader concept than the health-related quality of life?
  8. Line 134: Could you provide some examples of “peer factors”?
  9. Lines 135-136: Could you provide more explanations on structural equation modeling method? In fact, some readers may not be familiar with this method.

Tables, Expected Results and Discussion:

  1. Table 1 (lines 146-148): It would be better to make a figure rather than a table to describe the different stages of the project.
  2. Line 157: I would advise you to prefer the terms COVID-19 (for the disease) and SARS-CoV-2 virus (for the virus) which are the WHO recommended terms now.
  3. Line 170: Could you define the term “m-Health” ?

Author Response

Thank you very much for the peer comments. We addressed those one by one with explanation. Revised sentences in the manuscript are shown in italic with line number for your convenience.

Reviewer 1: General comments:

It would be useful to do a systematic review of the literature on this topic or at least a scoping review prior to your study.

Authors’ reply: We included a year plan for reviewing the literature. We have been doing it for a few months. In addition, we invited international researchers to discuss the topics around the research question. It has been ten times such a meeting has been conducted so far. DIHAC researchers published the report of such meetings and reviewed them regularly on the DIHAC study websites too (https://digital-ageing.com/surveys-and-blog/).

Introduction:

Lines 68-70: Could you confirm the meaning of this sentence? I have the impression that there is a misunderstanding.

Authors’ reply:  The sentence “Baby boomers, the older generation did not have to learn to use digital technology and media unlike in modern education for the Millennium generation.” Here in below is the meaning of the sentence.

The sentence is revised as

Unlike the Millennium generation, baby boomers, the older generation did not get the opportunity to learn to use digital technology and media in basic education.” (Line 70 to 71)

People who are born in 1945 did not have to learn computer and digital technology basic when they were going to primary schools or junior high schools. That is a big difference between the baby boomer generation and the millennial generation, those born after 2000.

Lines 113-120: Could you rephrase the objective sentence to select only one of the objectives as the primary objective and define the others as secondary objectives?

Authors’ reply: The paragraph is revised as

“The main objective of this study is to identify contextual influences of digital inclusion and its consequences on healthy ageing in Japan, ROK, Singapore and Thailand. The specific objectives of this research are (1) to measure and comparatively assess the digital skill of the older persons across four countries, (2) to measure and compare the e-health literacy of older persons in four countries, (3) to identify causal and sequential factors of digital inclusion among older persons such as personal characteristic, technical characteristic and positional characteristic (4) to identify paths and association between digital inclusion and older persons’ participation in health promotion and health-related quality of life (5) to exchange the contextually grounded models leading to digital inclusion and its impact on senior’s healthy ageing in four countries.” (Line 116 to 125)

Methods:

Lines 123-125: Could you provide more information on the sample size calculation?

Authors’ reply:   We provided the detailed sample size calculation as

“In each country, older participants will be recruited in a city which has ongoing health promotion activity for the seniors. Instrument:  We estimated the sample size for quan-titative to have a power of 80% and 95% confidence interval for the finding. Estimated samples constitute 360 participants for surveys in Japan, 360 in ROK and 480 in Thailand, and 360 in Singapore. Sample size calculation applied one sample estimation of proportion in STATA, based on the percentage of internet use among the older people We aimed to get sample sufficient to analyse small town models applying structural equation modelling. In addition, we also checked to secure the sample to exceed ten times of the factors in the instruments. number for each factor Generally the number of Each sample was inflated 20 percent for compensating the non-respond.”  (Line 129 to 139)

Lines 125-127: The inclusion criteria of the patients (age, cognitive ability, visual and auditory ability,...) should be explicitly mentioned.

Authors’ reply: We agree that the eligibility criteria are important. Hence, we elaborated as

“The inclusion criteria of the study participants will be the older residents (aged 65 years and over) of the community which has ongoing community-based health promotion activities for seniors, both male and female will be included in the study. For the participants with difficulty in responding, proxy respondents are also accepted.” (Line 140 to 143)

Lines 127-128: Could you provide the reference of the “Digital skill measurement scale” questionnaire and a copy of this questionnaire in the appendix?

Authors’ reply: Thank you very much. It is a published document. Therefore, we added as a reference (Reference number is 23).

Lines 128-129: Could you define the "eHealth Literacy" concept?

Authors’ reply: Thank you very much. We added the following definition of eHealth Literacy.

“Norman and Skinner defined the concept of eHealth literacy as the ability to seek, find, understand, and appraise health information from electronic sources and apply the knowledge gained to addressing or solving a health problem[23]. The participants’ perceived skill at using information and communication technologies for health will be assessed using 8-itemed 5-pointed Likert skill eHealth Literacy Scale[24]. “(line 164 to 168 )

Lines 128-129: Could you provide a copy of the “eHEALS eHealth Literacy Scale” questionnaire in the appendix?

Authors’ reply: Thank you very much. It is a published document. Therefore, we added as a reference (Reference number is 25).

Lines 130-132: Could you provide a copy of the “ICECAP-O (ICEpop CAPability measure for Older people)” questionnaire in the appendix?

Authors’ reply: Thank you very much. It is a published document. Therefore, we added as a reference (Reference number is 29).

Lines 130-133: Could you give examples to illustrate that the capability reflects a broader concept than the health-related quality of life?

Authors’ reply: The scale measuring the quality of life is different for community residents and dependent persons. In our sample, participants are community residents who has relatively higher level of autonomy and independence. Therefore quality of life based on capability concepts is more appropriate than other health related quality of life measurement .

“According to Sen’s capability approach in measuring the quality of life, measuring the quality of life should not be only based on perceived satisfaction but also include the conditions that allow or enable oneself to do things or be that one values [25]. Capability also differs according to an individual's characteristics and the social environment that they interact with. Therefore, we will use ICECAP-O Instrument for older people to measure the capability well-being of study participants. It is a 5-items tool that measures capability, the extent a person can do things that they valued [26], and reflects the general quality of life in a broader sense[27]. The domains of ICECAP-O, based on the capability concept, include 1) attachment, 2) security, 3) role,4) enjoyment, and 5) control. The do-mains reflect the essence of healthy and active ageing. It coincides with the study's aim to explore the effect and sequences of using digital technology on the healthy ageing of the older population.” (178 to 190)

Line 134: Could you provide some examples of “peer factors”?

Authors’ reply:

With the physical distancing rules under effect during the Covid-19 pandemic, older people started to use social media and video calling applications to stay connected with friends and family. The use of these applications by one older person could influence the other to maintain the social connection. Peer can trigger the use of digital devices and technologies and also help to sustain. In addition, peer support programs among senior citizens have been used and shown effectiveness in promoting physical activities and increasing quality of life. The effect of peer factors on digital technology usage is still needed to be explored.

Lines 135-136: Could you provide more explanations on structural equation modeling method? In fact, some readers may not be familiar with this method.

 Authors’ reply: Thank you for your advice. We explained the structural equation modelling (SEM) as

“Structural equation modelling (SEM) method will be used to detect pathways from explanatory variables to the outcome variables, constructing the conceptual model or mechanism. It identifies the unobserved variables. It can reveal the direct and indirect association. Furthermore, it allows us to see not only the causal pathways but also con-sequential pathways. After we developed the SEM models, model fit indexes will be computed.” (Line 233 to 238)

Tables, Expected Results, and Discussion:

Table 1 (lines 146-148): It would be better to make a figure rather than a table to describe the different stages of the project.

Authors’ reply: Thank you for your suggestion. Since this paper is the research protocol and we have not yet finalized the final expected conceptual model, therefore, we think it might be a good way to use a table to elaborate our research plan with a timeline and detailed content of the research in each phase.  After the data collection, and analysis and we developed the conceptual model of the research, we will take your suggestion and use a figure to show our research result.

Line 157: I would advise you to prefer the terms COVID-19 (for the disease) and SARS-CoV-2 virus (for the virus) which are the WHO recommended terms now.

Authors’ reply: We have checked the related words and carefully replaced respectively (Line 252).

Line 170: Could you define the term “m-Health”

Authors’ reply: WHO m-health definition has been added with a reference (reference number 33).

“The Global Observatory for eHealth defined mobile health(mHealth) as medical and public health practice supported by mobile devices, such as mobile phones, patient monitoring devices, personal digital assistants (PDAs), and other wireless devices.” (Line 267 to 269)

Reviewer 2 Report

The authors designed research protocol to study cross-cultural influence of digital in Japan and other 3 asian countries. The idea is novel and significant, since the aging population is becoming more and more now and thus increasing society burden. Therefore, improving the life of aging population is critical to whole society and the world. The aims of this research is clear, however, the method is not detailed. 

  1. selection of samples: what is criteria of inclusion of aged people? 360 people in each country is enough or not?
  2. The educational levels in four countries should vary a lot. Japan and Singapore are developed countries but Tailand is not, has this factor been considered into the design of research? For aging population, the levels of education is discontinued or absence during their young age due to poverty or war, which may affect the preference of using digital device when they are aged.

Author Response

The authors designed research protocol to study cross-cultural influence of digital in Japan and other 3 asian countries. The idea is novel and significant, since the aging population is becoming more and more now and thus increasing society burden. Therefore, improving the life of aging population is critical to whole society and the world. The aims of this research are clear, however, the method is not detailed. 

Thank you very much for the peer comments. We addressed those, one by one with the explanation. Revised sentences in the manuscript are shown in italic with the line number for your convenience. In the new version of the manuscript, we re-structured the research method part (From line 126 to line 246) with more detail, explaining objectives in each phase, every instrument and plan of analyses.

  1. selection of samples: what is criteria of inclusion of aged people? 360 people in each country is enough or not?

Authors’ reply: The inclusion criteria of the study participants and sample size calculation is explained as

2.1 Study Design and Sampling (Line 127 to 143)

“This is a five-year project. A mixed-method study will apply an explanatory sequential (QUAN → qual), two-phase research design [22]. In each country, older participants will be recruited in a city which has ongoing health promotion activity for the seniors. Instrument:  We estimated the sample size for quantitative to have a power of 80% and 95% confidence interval for the finding. Estimated samples constitute 360 participants for surveys in Japan, 360 in ROK and 480 in Thailand, and 360 in Singapore. Sample size calculation applied one sample estimation of proportion in STATA, based on the percentage of internet use among the older people We aimed to get sample sufficient to analyze small town models applying structural equation modelling. In addition, we also checked to secure the sample to exceed ten times of the factors in the instruments. number for each factor Generally the number of Each sample was inflated 20 percent for compensating the non-respond.

The inclusion criteria of the study participants will be the older residents (aged 65 years and over) of the community which has ongoing community-based health promotion activities for seniors, both male and female will be included in the study. For the participants with difficulty in responding, proxy respondents are also accepted.”

  1. The educational levels in four countries should vary a lot. Japan and Singapore are developed countries but Tailand is not, has this factor been considered into the design of research? For aging population, the levels of education is discontinued or absence during their young age due to poverty or war, which may affect the preference of using digital device when they are aged.

Authors’ reply: Thank you very much for your comment. The educational background of the seniors varies between these four study countries. And we have included educational level in the questionnaire and will measure the school education achievements of the study participants.

Reviewer 3 Report

The start of this manuscripts promised more than later on has been delivered. It appears to be a protocol of a study without real results yet. I'm in favor of the publication of a research protocol, However I'm not sure if IJERPH is the right platform for this. Furthermore it should be stated very cleary from the start of the manuscript that this is a protocol and not an executed study. 

If it is a protocol the method section needs major revisions. The two phases should be described in much more detail. The research question in each phase and study should be elaborated. The sample size calculation is incomplete. It is stated that a power of 80% is the aim, However it is unclear on what the calculation is based on. In this way it is impossible to check the calculation. Who is eligible to participate, how will the participants be recruited, do they sign informed consent..... etc. Also the statistical analysis should be described in detail. 

Author Response

The start of this manuscripts promised more than later on has been delivered. It appears to be a protocol of a study without real results yet. I'm in favor of the publication of a research protocol, However I'm not sure if IJERPH is the right platform for this. Furthermore it should be stated very cleary from the start of the manuscript that this is a protocol and not an executed study. 

If it is a protocol the method section needs major revisions.

The two phases should be described in much more detail. The research question in each phase and study should be elaborated.

Authors’ reply: Thank you very much for the peer comments. We addressed those, one by one with the explanation. Revised sentences in the manuscript are shown in italic with the line number for your convenience. In the new version of the manuscript, we re-structured the research method part (From line 126 to line 245) with more detail, explaining objectives in each phase, every instrument and plan of analyses.

“2. Research Methods (Line 126)

2.1 Study Design and Sampling (Line 127 to line 143)

This is a five-year project. A mixed-method study will apply an explanatory sequential (QUAN → qual), two-phase research design [22]. In each country, older participants will be recruited in a city which has ongoing health promotion activity for the seniors. Instrument:  We estimated the sample size for quantitative to have a power of 80% and 95% confidence interval for the finding. Estimated samples constitute 360 participants for surveys in Japan, 360 in ROK and 480 in Thailand, and 360 in Singapore. Sample size calculation applied one sample estimation of proportion in STATA, based on the percentage of internet use among the older people We aimed to get sample sufficient to analyse small town models applying structural equation modelling. In addition, we also checked to secure the sample to exceed ten times of the factors in the instruments. number for each factor Generally the number of Each sample was inflated 20 percent for compensating the non-respond.

The inclusion criteria of the study participants will be the older residents (aged 65 years and over) of the community which has ongoing community-based health promotion activities for seniors, both male and female will be included in the study. For the participants with difficulty in responding, proxy respondents are also accepted.

2.2 Study Phases and Research Instruments (Line 144)

2.2.1 Phase One: Quantitative Survey (Line 145 to line 152)

Objectives of the study phase one comprise (1) to measure and comparatively assess the digital skill of the older persons across four countries, (2) to measure and compare the e-health literacy of older persons in four countries, (3) to identify causal and sequential factors of digital inclusion among older persons such as personal characteristic, technical characteristic and positional characteristic  (4) to identify paths and association between digital inclusion and older persons’ participation in health promotion and health-related quality of life

2.2.1.1 Research Instruments: (Line 153 to 192)

Translation of instrument will follow forward translation, backtranslation, cognitive test and pilot study to check reliability and retain the validity. The research instruments that will be used in this study are as below:

1) Digital Skill Measurement Scale (Line 157 to 162)

The perceived digital skill of the study participants will be measured using 20-itemed “Digital skill measurement scale” of London School of Economics, transcultural translated version [23]. It is 23-itemed, 5-pointed Likert scale measuring five do-mains: operational internet skill, information navigation, communicational/social internet skill, creative skill, mobile internet skill.

2) eHealth Literacy Scale (Line 163 to 168)

Norman and Skinner defined the concept of eHealth literacy as the ability to seek, find, understand, and appraise health information from electronic sources and apply the knowledge gained to addressing or solving a health problem [24]. The participants’ perceived skill at using information and communication technologies for health will be assessed using 8-itemed 5-pointed Likert skill eHealth Literacy Scale [25].

3) Health Promotion Activity Participation Registers or Record (Line 169 to 173)

The study participants will be selected in the community that has ongoing health promotion activities for older adults. Therefore, the registration record will be obtained with the permission of the participants and activity organizers. Type and frequency of the activity will be specified.

4) ICECAP-O (ICEpop CAPability measure for Older people) (Line 174 to 189)

In this study, we focus on community-dwelling older adults and their participation in health promotion activities. In measuring the quality of life, health is an important domain but not the only one. Capability measures the ability to do valuable things or be with the given resources. According to Sen’s capability approach in measuring the quality of life, measuring the quality of life should not be only based on perceived satisfaction but also include the conditions that allow or enable oneself to do things or be that one values [26]. Capability also differs according to an individual's characteristics and the social environment that they interact with. Therefore, we will use ICECAP-O Instrument for older people to measure the capability well-being of study participants. It is a 5-items tool that measures capability, the extent a person can do things that they valued [27], and reflects the general quality of life in a broader sense [28]. The domains of ICECAP-O, based on the capability concept, include 1) attachment, 2) security, 3) role,4) enjoyment, and 5) control. [29] The domains reflect the essence of healthy and active ageing. It coincides with the study's aim to explore the effect and sequences of using digital technology on the healthy ageing of the older population.

5) Personal characteristics, technical characteristics and positional characteristics, family factors, peer factors, community factors will be investigated applying resource and appropriation theory in digital sociology [11]. (Line 190 to 192)

2.2.2 Phase Two: Qualitative Inquiries (Line 195 to 202)

The second phase of this study is qualitative interviews, which aim to identify the cultural pattern, and contextual factors making sense of what older persons in different countries materialize, say, do, think, and feel to reveal deeper beliefs and core values about digital inclusion and healthy ageing.

Moreover, we seek to exchange the contextually grounded models leading to digital inclusion and its impact on senior’s healthy ageing in four countries.

At least five focus group interviews inviting older persons, family caregiver and multiple stakeholders, will be conducted at each site.

2.2.2.1 Research Instrument: (Line 203 to 212)

Qualitative thematic analysis will identify the cultural pattern, and contextual fac-tors making sense of what older persons in different countries materialize, say, do, think, and feel to reveal deeper beliefs and core values about digital inclusion and healthy ageing, applying a cross-cultural empathy framework [30] (Table 1).

Findings from the two phases will be critically reflected and integrated to identify the DIHAC model in each country and for Asia in the study year 4. (Table1) Applying a cross-cultural empathy framework, this explanatory sequential two-phase research may chart the Asia DIHAC model, to strategize the empowerment of communities and individuals for healthy ageing communities.

2.3 Ethical Consideration (Line 218 to 224)

The Juntendo University Ethics Committee has reviewed and approved the protocol of Digitally inclusive, healthy aging communities (DIHAC): A cross-cultural study in Japan, Republic of Korea, Singapore and Thailand (DIHAC) study. Approval Number is E22-0057-M01.

Written informed consent will be obtained from all participants, and the purpose of this study will be carefully explained to them after their permission.

2.4 Data Analyses (Line 225 to 245)

STATA version SE 17 (Stata Corp 4905, Lakeway Drive, College Station, Texas 77845 USA) will be applied for data analysis. We will use descriptive analysis to elaborate the sociodemographic information of study objectives. Statistical significance of various factors will be measured through bivariate analysis (Chi-square test). The Association of dependent and independent variables will be analysed via multivariate logistic regression model. A p-value of less than 0.2 in the bivariate analysis will be certified the factors to be selected in the multivariate logistic regression analysis. Structural equation model-ling (SEM) method will be used to detect pathways from explanatory variables to the outcome variables, constructing the conceptual model or mechanism. It identifies the unobserved variables. It can reveal the direct and indirect association. Furthermore, it allows us to see not only the causal pathways but also consequential pathways. After we developed the SEM models, model fit indexes will be computed. Statistical significance defined as a p-value of less than 0.05 with a 95% confidence interval (CI).

For the qualitative study analysis, we will use a cross-cultural empathy framework to carry out inquires and analyze the collected data. Literature review to understand the development of digital policies and healthy ageing programs in four countries, to understand the systems and contextual factors will precede the data collection and analysis. Cross-cultural exchange in the results and model aims to identify how digital policies are being adopted by the older persons, facilitators, barriers in each setting and finally the empowerment models.”

The sample size calculation is incomplete. It is stated that a power of 80% is the aim, however it is unclear what the calculation is based on. In this way it is impossible to check the calculation.

 Who is eligible to participate, how will the participants be recruited, do they sign informed consent..... etc.

Authors’ reply: Thank you for your comment. The explanation is included above in the 2.1.1 Study Design and Sampling session.   

Also the statistical analysis should be described in detail. 

Authors’ reply: Thank you for your comment. The explanation is included above in the 2.4 Data analysis session.

Round 2

Reviewer 1 Report

Title:

Could you explicitly state in the title that this is a study protocol and not a study?

Methods:

Lines 128-138: About the sample size calculation, could you provide the values of the statistics you used (percentage of internet use among the older people, type and number of factors considered,...) so that we can do this calculation again ?

Author Response

Thank you very much for the review and comments.

Reviewer's comment :

Title:

Could you explicitly state in the title that this is a study protocol and not a study?

Author's response and revision: We revised the title.

Digitally inclusive, healthy ageing communities (DIHAC): A study protocol of a cross-cultural study in Japan, Republic of Korea, Singapore, and Thailand

Methods:

Lines 128-138: About the sample size calculation, could you provide the values of the statistics you used (percentage of internet use among the older people, type and number of factors considered,...) so that we can do this calculation again ?

Authors' response and revision Line 138-144

For instance, internet use among the older people in Japan has been reported as 50% (null), but we estimated as 40% (alternative) and we applied the command for one sample estimation in STATA 16 SE. We aimed to get sample sufficient to analyze small town models applying structural equation modelling. In addition, we also checked to secure the sample to exceed ten times of the factors, less than 30, in the instruments. number for each factor Generally the number of Each sample was inflated 20 percent for compensating the non-respond.

 We thank you very much for guiding us to completely describe how we estimated the sample. command are rather different in different version of STATA. such as power one proportion 0.50 0.4, n(300)

Reviewer 3 Report

Thank you very much for your efforts to improve this manuscript. My compliments for the thorough way you have addressed my main concerns. 

Although in IJERPH it is custom to mention in the left corner above the title ‘study protocol’ I would prefer to mention this also in the title itself as after publication in case of citation only the original title will be visible. My suggestion is to change the title into:

Digitally inclusive, healthy ageing communities (DIHAC): A study protocol of a cross-cultural study in Japan, Republic of Korea, Singapore, and Thailand 

Furthermore ‘a study protocol’ should be added to the abstract and it would also be sensible to add this at the end of the introduction, after line 115. For example you state in line 113-115 the following;

Our research questions are: (1) what are the contextual influences for digital inclusion? (2) what are its  consequences on healthy ageing in Japan, South Korea, Singapore, and Thailand? 

è You can add after this sentence “To answer these research questions we developed a study protocol”

The method section has been improved considerably. I have still some minor issues.

You mention in line 143 “For the participants with difficulty in responding, proxy respondents are also accepted.” ïƒ  I think this can cause some interpretation problems and potential over- or underestimation of the results. If you want to include people with limited literacy or illiterate or people with cognitive impairments then I would suggest to stratify in the analyses and not to include them in the general population.

The assessment instruments are well described. Are the assessment instruments administered only once or is there also some follow-up assessment ? in other words is this a cross sectional design ? Please elaborate. 

Although the mixed method as described (and here it is my assumption that the quantitative part is a cross sectional design)  will give insight in the digital inclusiveness of the investigated population it will not allow to gain insight in causal relationships as mentions in line 237 of the Statistical analysis section. 

Author Response

Thank you very much for your efforts to improve this manuscript. My compliments for the thorough way you have addressed my main concerns. 

Reviewer: Although in IJERPH it is custom to mention in the left corner above the title ‘study protocol’ I would prefer to mention this also in the title itself as after publication in case of citation only the original title will be visible. My suggestion is to change the title into:

Digitally inclusive, healthy ageing communities (DIHAC): A study protocol of a cross-cultural study in Japan, Republic of Korea, Singapore, and Thailand 

Author response and revision: We totally agreed with you and revised the title as

Digitally inclusive, healthy ageing communities (DIHAC): A study protocol of a cross-cultural study in Japan, Republic of Korea, Singapore, and Thailand 

Reviewer's comment : 

Furthermore ‘a study protocol’ should be added to the abstract and it would also be sensible to add this at the end of the introduction, after line 115. For example you state in line 113-115 the following;

Our research questions are: (1) what are the contextual influences for digital inclusion? (2) what are its  consequences on healthy ageing in Japan, South Korea, Singapore, and Thailand? 

Author response and revision

We added a sentence in the abstract :

"Logics and methods from this protocol would be useful to replicate the study in many countries globally."

Line 115: We added a sentence at the end of introduction.

To answer these research questions, we developed a study protocol and shared the logics and methods for further studies globally.

Our research questions are: (1) what are the contextual influences for digital inclusion? (2) what are its  consequences on healthy ageing in Japan, South Korea, Singapore, and Thailand? 

for is added.

Reviewer's comment: 

The method section has been improved considerably. I have still some minor issues.

You mention in line 143 “For the participants with difficulty in responding, proxy respondents are also accepted.” à I think this can cause some interpretation problems and potential over- or underestimation of the results. If you want to include people with limited literacy or illiterate or people with cognitive impairments, then I would suggest to stratify in the analyses and not to include them in the general population.

Author's response: We deleted the sentence. 

Reviewer's comment:

The assessment instruments are well described. Are the assessment instruments administered only once or is there also some follow-up assessment ? in other words is this a cross sectional design ? Please elaborate. 

Although the mixed method as described (and here it is my assumption that the quantitative part is a cross sectional design)  will give insight in the digital inclusiveness of the investigated population it will not allow to gain insight in causal relationships as mentions in line 237 of the Statistical analysis section. 

Author's response: Thank you very much.

We would like to explain this to the reviewer. We are not trying to say cause and effect association in a cross-section study. It has been known that internet use and digital inclusion has positive impacts on health.

We will use a mixed method study.  We are not trying to establish a relationship between an explanatory variable (X1) and some outcome (Y), controlling for other factors (X2), but researchers want to better understand how X1 generates Y. (ref) 

(ref 31) Weller, Nicholas; Barnes, Jeb. Finding Pathways: Mixed-Method Research for Studying Causal Mechanisms (Strategies for Social Inquiry) . Cambridge University Press. 

We carefully chose the word causal pathway and sequential pathway. It means the model. To identify the model in such an approach is well recognized method. It is to be clear that we did not use the word “causal association”.

We added a new reference 31 

" Weller, Nicholas; Barnes, Jeb. Finding Pathways: Mixed-Method Research for Studying Causal Mechanisms (Strategies for Social Inquiry) . Cambridge University Press.”